# Agro-Dealers' Knowledge, Perception, and Willingness to Stock a Fungal-Based Biopesticide (ICIPE 20) for Management of *Tuta absoluta* in Kenya

Francis Ogutu [1,2], Beatrice W. Muriithi [2,*], Patience M. Mshenga [1], Fathiya M. Khamis [2], Samira A. Mohamed [2] and Shepard Ndlela [2]

1   Department of Agricultural and Agribusiness Management, Egerton University,
    Egerton-Njoro P.O. Box 536-20115, Kenya; fogutu@icipe.org (F.O.); pmshenga@egerton.ac.ke (P.M.M.)
2   International Centre of Insect Physiology and Ecology (ICIPE), Nairobi P.O. Box 30772-00100, Kenya;
    fkhamis@icipe.org (F.M.K.); sfaris@icipe.org (S.A.M.); sndlela@icipe.org (S.N.)
*   Correspondence: bmuriithi@icipe.org

**Abstract:** In sub-Saharan Africa (SSA), tomato is an economically important crop that contributes not only to employment and income, but also food security. Like the rest of the SSA countries, tomato production in Kenya is constrained mainly by pests and diseases, key among them being the tomato leaf miner (*Tuta absoluta*), which can cause 80–100% losses if not properly managed. To suppress this pest, the International Centre of Insect Physiology and Ecology (ICIPE) and partners are introducing a fungal-based biopesticide (ICIPE 20) in an Integrated Pest Management (IPM) approach as a sustainable alternative to the sole use of synthetic pesticides. This study was carried out before the introduction of the biopesticide to assess its commercial feasibility among agro-dealers, using Kirinyaga County in Kenya where tomato production is predominant, as the study area. Specifically, the study assessed the knowledge, perception, and willingness to stock biopesticide using a market survey involving 141 agro-dealers. Successful commercialization of a new product is assumed to be the cumulative result of traders' and buyers' knowledge and perceptions about the product. The results show that a higher proportion of agro-dealers were willing to pay for ICIPE 20 at a higher price than Coragen®, the most popular insecticide for management of *T. absoluta*. The regression analysis revealed that individual characteristics such as age, education, access to social networks and credit facilities, and information are correlated to the agro-dealer's knowledge, perception, and willingness to stock the biopesticide. Training agro-dealers may promote greater uptake of the biopesticides through enhancing their knowledge and perception towards the effectiveness of the product.

**Keywords:** integrated pest management; *Tuta absoluta*; fungal-based biopesticide; willingness to stock; Kenya; Africa

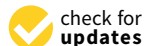



## 1. Introduction

Tomato (*Solanum lycopersicum* L.) is an economically important and nutritious vegetable that is extensively cultivated in the horticultural sub-sector in sub-Saharan Africa [1]. The crop plays an important role in the provision of food and nutrition, especially among impoverished rural communities. Additionally, the tomatoes reach urban markets through traders and middlemen thus generating income to various actors along the value chain. The surplus produce is often exported to regional and international markets thereby earning the country much-needed foreign currency. Kenya is among the leaders in tomato production in Africa, with an average annual production of approximately 410,033 metric tons [2]. Kirinyaga, Kajiado, and Narok counties are the major tomato-producing regions, accounting for 37% of the total tomato output of the country. Despite this impressive performance, tomato yields have remained below the country's potential due a myriad of impediments,

including pests and diseases, key among them being the invasive South American tomato leaf miner (*Tuta absoluta*, Lepidoptera: Gelechiidae).

*Tuta absoluta* is the most economically important pest in tomato production, responsible for 80–100% production losses if not managed appropriately. Since the first appearance of *T. absoluta* in Kenya in 2013, tomato growers have chiefly depended on the use of synthetic chemical pesticides to control the pest. However, pesticides are known to cause unprecedented negative effects, including ecological damage [3]. They also cause serious health hazards among workers during their manufacture, formulation, handling at the sales point, and field application in crop protection [4]. High pesticide residues in food substances, which are harmful to human health and the environment, is attributed to over(mis) use of synthetic chemicals [5]. *T. absoluta* has a high propensity to develop resistance and is currently known to have developed resistance to several major classes of insecticides [6,7].

As a result of the negative effects of synthetic chemical pesticides, there has been an increasing demand for organic food products [8]. Organic farming combines both indigenous practices and science to produce crops that flourish in the absence or minimal use of synthetic insecticides and herbicides. It also incorporates integrated pest management components such as mass trapping of insect pests, orchard sanitation, and biological control [9]. Biological control (BC) involves the use of biopesticides or/and living organisms such as parasitoids or a beneficial insect that destroys the harmful one [10]. In this regard, the International Centre of Insect Physiology and Ecology (ICIPE) and partners are introducing an ecologically and economically friendly fungal-based biopesticide, dubbed ICIPE 20, for sustainable management of *T. absoluta* in Africa. This is in addition to the already introduced arsenal which includes the use of the effective parasitoid *Dolichogenidea gelechiidivoris* (Hymenoptera: Braconidae), imported from Peru. ICIPE 20 is an entomopathogenic fungus: *Metarhizium anisopliae* Sorokin (1883) isolated from soil in Migori county in Kenya, in 1989. The isolate has been demonstrated to cause between 82–92% mortality in *T. absoluta* larvae and adult moths and is highly compatible with the pheromone lures currently being used for monitoring and mass trapping moths [11]. The isolate if commercialized as a biopesticide will form an integral component of *T. absoluta* Integrated Pest Management (IPM) as it can be sprayed on the ground, and on the crop to kill both larvae and adult moths of the pest. Besides, the biopesticide can also be used in auto dissemination devices in conjunction with lures, which are commercially available in the market.

Although the use of biopesticides exhibits potential economic, health, and environmental benefits as demonstrated in related studies [12], wide-scale commercialization and adoption of the technology will depend on agro-dealers' and farmers' knowledge, perceptions, and the agro-dealer's willingness to stock the new technology [13] Conducted before scaling out the biopesticide, the objective of this study was to assess the knowledge and perceptions and willingness to stock the fungal-based biopesticide (ICIPE 20) among key agro-dealers who are direct retailers of the tomato producers, using Kirinyaga County as the case study. The study utilized the contingent valuation (CV) method to elicit agro-dealers' willingness to stock the biopesticide. This is the first study to assess the willingness to stock the effective and eco-friendly ICIPE 20 fungal biopesticide among agro-dealers. This study complements previous literature by critical analysis of potential factors that are likely to promote marketing of the relatively new biopesticide for management of the invasive tomato infesting leaf miner. Wider adoption of the biopesticide will depend on the trader's awareness, perception, and willingness to stock and sell the product within the farmer's reach. The importance of the knowledge, attitudes, and perceptions in commercialization and adoption of pest management technologies in developing countries is underscored in previous literature [13,14]

The results show that traders stocked a wide range of chemicals for the management of *T. absoluta* with the most used pesticide being Coragen® (active ingredient chlorantraniliprole). A higher proportion of agro-dealers (82%) was willing to pay for ICIPE 20 at the same price as Coragen® and most importantly, at a much higher price of approximately

Ksh 750 (US$7.5) per liter. The regression analysis revealed that individual characteristics such as age, education, access to social networks, access to credit facilities, and information are correlated to the agro-dealers' knowledge, perception, and willingness to stock a fungal biopesticide. The remainder of the paper is organized into three sections. Section 2 outlines the conceptual framework, and presents the materials and methods, Section 3 provides the results and discussions, while Section 4 presents the conclusions and provides policy implications.

## 2. Materials and Methods

### 2.1. Conceptual Framework

This study is based on the theory of profit maximization which assumes that traders are compelled to search for new technologies to innovate efficient and effective products, reduce costs of production and increase revenues. However, these products may not have readily available markets thus making it impractical to determine their profitability. Determining the feasibility of stocking these new products is critical to the traders since they are profit-oriented. A bidding game in the contingent valuation method (CVM) was used in this study. It involves a series of yes/no questions aimed at determining the maximum willingness to stock for new products by agro-dealers. This approach is used in the valuation of non-market products for economic value as illustrated by the study conducted by Chege and Groot [15] and further expounded by [16]. The repeated nature of questions in the bidding game gives the respondent enough time to make a decision based on the maximum favourable price that he or she is willing to pay for the product.

The CVM approach employs three methods, namely single-bounded, double-bonded and multi-bounded models. In the single-bounded model, the respondent is only offered one bid to either accept or reject. According to Carson [17], the method is not appropriate since it requires a large sample size and is not statistically efficient. Studies have illustrated the effectiveness of the double-bounded model since it incorporates more information about the respondents' willingness to pay (WTP), and therefore providing efficient estimates [11,18,19]. The model offers a second bid, either higher or lower depending on the respondent's first response, unlike the single-bounded model [17]. The multiple-bounded model allows for multiple bids and choices, which offers the possibility of including alternatives for uncertainty. However, the multiple-bounded models are design-biased and influenced by the range of bids included [20]. Because of the endogeneity of the responses, it is common for a single participant to send a significant number of responses, which can make data analysis difficult. Bateman et al. [21] noted various concerns when eliciting willingness to pay through repeated constrained questions, including boredom, quality deterioration, outrage, free riding, and yea-saying. Participants may become weary as a result of the large number of questions asked in a short period of time and therefore provide bias results especially to later responses and thus subjective predicted WTP [22].

This study employed the double-bounded model which has a good theoretical justification and provides unbiased estimates [23]. In this model, the respondents are offered a two-sequence bid offer. They are asked if they will accept or reject the bid, then the next bid is asked depending on the respondent's initial response. Depending on the type of response from the respondent of either yes/no, a higher bid is offered if the respondent said 'yes' and a lower bid offered if the respondent said 'no'. The outcome presents four possible responses which include (i) Either both responses are 'yes', (ii) both of them are 'no', (iii) a 'no' followed by a 'yes' (iv), or a 'yes' followed by a 'no'. An assumption prevails that when the maximum willingness to pay is less than or equal to the lowest bid $\left( maxWTP \leq Bid_j^L \right)$ then the first and second bid offers are rejected by the respondent. The maximum willingness to pay therefore lies between the lower and the first bid offer $\left( Bid_j^L \leq maxWTP < Bid_j \right)$, if the respondent rejects the first bid offer but accepts the second lower bid. If the respondent is willing to accept the first bid but rejects the second higher bid offer, then an assumption prevails that the respondent's maximum willingness to pay lies between the second higher bid and the first bid offer $\left( Bid_j^H > maxWTP > Bid_j \right)$.

Finally, if the respondent is willing to accept both the first and the second higher bids, then an assumption prevails that the respondent's maximum willingness to pay is greater than or equal to the second higher bid $\left( maxWTP \geq Bid_j^H \right)$.

For this study, the agro-dealers were first asked about the most demanded pesticide for the management of the leaf miner, and the price per litre. This price was offered as the first bid for the fungal-based biopesticide. Those who answered positive (yes), were offered a higher price, which was increased by the randomly assigned premium (i.e., 5%, 10%, 20%, 30%, or 50%). The respondents who answered 'no' to the first bid were offered a lower price similarly reduced by a randomly assigned discount (i.e., 5%, 10%, 20%, 30%, 50%). The binary probit model was then employed to analyse the different factors influencing agro-dealers' willingness to stock a fungal-based biopesticide.

### 2.2. Study Area and Data Collection

The data were collected from 141 agro-dealers in Kirinyaga County in Kenya. The study site and respondents were selected through the multi-stage sampling technique. In the first stage, Kirinyaga County was purposively selected since it is the leading tomato-producing county in Kenya [24]. The county accounts for about 17% of the total country's production, followed by Kajiado (11.8%) and Taita Taveta (8.5%) [24]. In the second stage, two sub-counties, namely Mwea East and Mwea West were purposefully selected due to their predominance in the total production (Figure 1). A census of the agro-dealers within the two sub-counties was then developed with the help of the agricultural officers. All the listed agro-dealers (141) were targeted for the survey. Data were collected by well-trained enumerators who were post-graduate students with vast experience in and data collection. They were trained and supervised by the lead author of this study. Data collection was done using a pre-tested semi-structured questionnaire programmed in the CSPro version 7.0 software to reduce data collection errors. The questionnaire was written in English but comprehensively explained in the national language Swahili and the local language of the study area to the enumerators to ensure accurate translation of the questions during the actual survey. Additionally, to ensure relevant and accurate information from the respondents, the sub-county agricultural officers were engaged, to assist in mobilizing the agro-dealers and introduce the enumerators to them. The study area is shown below (Figure 1).

The information collected included the agro-dealers' socioeconomic characteristics such as age, gender, education level and the number of years in agro-vet business operation, their knowledge, and perception towards the use of biopesticides, their capacity to stock new biopesticides, their willingness to stock the fungal-based biopesticides for management of *Tuta absoluta*, and the costs and benefits associated with commercializing biopesticides. This study was approved by the *icipe*'s social science and impact assessment unit, in line with the institution's research ethics policy guidelines. The questionnaire had an introductory statement that sought the agro-dealers consent to participate in the survey. Thus, respondents participated freely and the confidentiality of personal information was guaranteed.

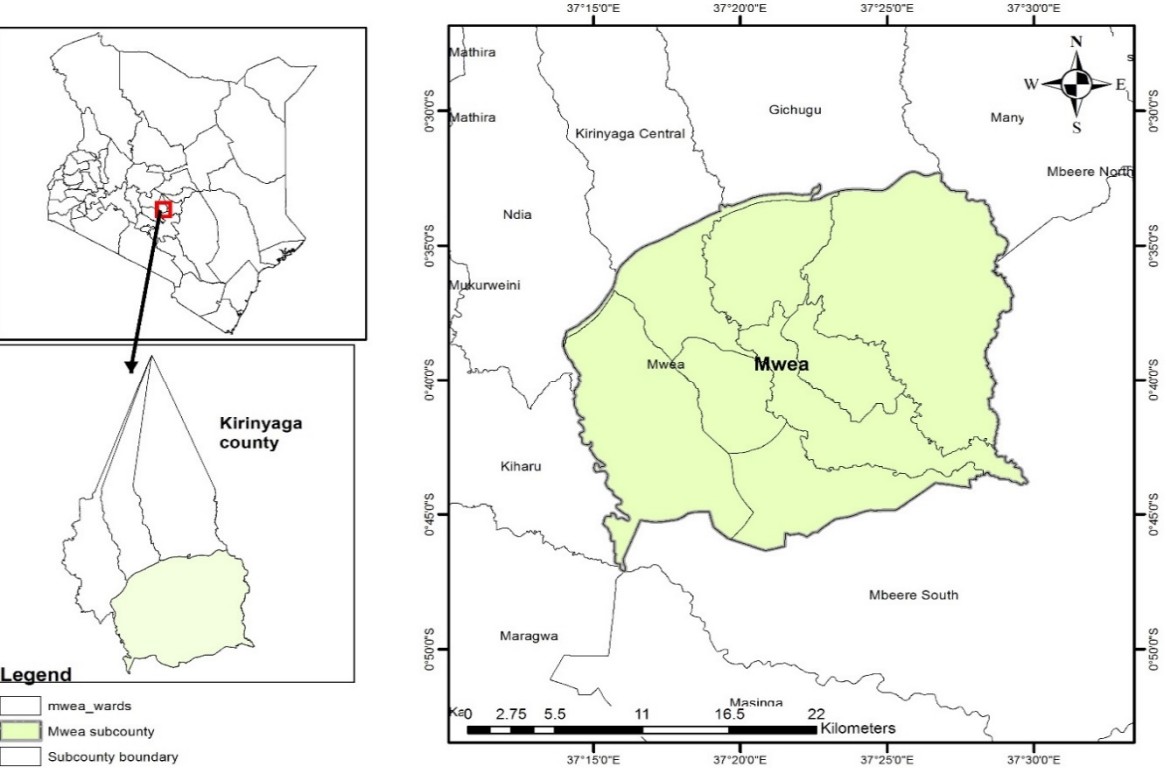

**Figure 1.** Map of Kenya showing the study area.

### 2.3. Empirical Model

The present study adopted a binary regression model to identify covariates that influence agro-dealers willingness to stock a fungal-based biopesticide. The model gives the probability that *y* (dependent variable) is equal to one, is chosen conditional on a set of independent variables. The most used approaches are the logit and probit models which assume that the functional form of dependence on the independent variables is known. Both the probit and logit models yield similar inferences though not identical since the logit regression has a standard logistic distribution of errors while the probit has a normal distribution of errors [25]. Therefore, the probit model is appropriate since it assumes normality that is; a mean of 0 and a standard deviation of 1.

The probit model is a statistical probability model with two categories in the dependent variable. Probit analysis is based on the cumulative normal probability distribution. The binary dependent variable, *y*, takes on the values of zero and one. For our study, the dependent variable of the model is the willingness to stock fungal-based biopesticide, which takes a value of 1 if Yes, and 0 otherwise. The probability *pi* of choosing any alternative over not choosing it can be expressed as follows:

$$pi = prob\left(Yi = \frac{1}{X}\right) = \int_{-\infty}^{Xi\beta} (2\pi) - \frac{1}{2}\exp\left(-\frac{t2}{2}\right)dt = \Phi\left(xi\ \beta\right) \tag{1}$$

where $\Phi$ represents the cumulative distribution of a standard normal random variable. Consequently, the factors influencing agro-dealers willingness to stock a fungal based biopesticide was estimated using the probit model of the following form:

$$Y_i = \beta_0 + \beta\prime X_i + e_i \tag{2}$$

where $X_i$ is a vector of characteristics of interest for agro-dealer *i* and their household, and $e_i$ is an error term. The selection of the independent variables (*X*) for this study was guided by the review of previous literature on willingness to pay for new products [14–16],

and the knowledge, attitudes, and practices towards pesticide use [17–19]. These include individual characteristics such as age and education, and business-level variables such as years of business experience, use of hired labor, access to information, social capital and networks, and managerial support. Further, perception studies such as Abudulai [26] and Schreinemachers [27] were also reviewed to provide an in-depth analysis for robust results.

## 3. Results and Discussion

### 3.1. Selected Socio-Economic Characteristics of the Study Population

The description and summary of selected socioeconomic characteristics of the sampled agro-dealers are presented in Table 1. On average, the agro-dealers were 35 years of age, with 13 years of schooling, implying that the majority of the traders were young and had attained a secondary level education. Education is considered as a human capital that facilitates the use of available information to reduce the existing constraints [28]. For example, agro-dealers who are better educated may have better business skills that enable them to operate their businesses more efficiently and therefore be more likely to stock the fungal-based biopesticide. The average number of years of business operation was 5 years, depicting a general low level of experience in the agro-vet business.

**Table 1.** Socio-economic characteristics of agro-dealers in Kirinyaga County (*n* = 141).

| Agro-Dealer Characteristics | Statistics |
|---|---|
| Age of the agro-dealers (years) | 35.73 (10.65) |
| Years of schooling of agro-dealers | 13.00 (2.13) |
| Years in Agro-vet operation | 4.97 (5.19) |
| Agro-dealer access to credit (1 = yes, 0 = No) (%yes) | 46.1 |
| Hired labour in the agro-vets (1 = yes, 0 = No) (%yes) | 32.62 |
| Agro-dealer access to information (1 = yes, 0 = No) (%yes) | 92.20 |
| Agro-dealer access to social networks (1 = yes, 0 = No) (%yes) | 43.97 |
| Managerial support by manufacturers and suppliers (1 = yes, 0 = No) (%yes) | 81.56 |

Note: Standard deviation in parenthesis.

About 46% of the agro-dealers stated that they had access to credit facilities. However, only 7% of them reported that they needed credit for the agro-vet business operations. Chia et al., [29] observed that access to credit enhances willingness to stock new products, with a corresponding positive perception towards them.

With respect to hired labour, the results indicated that 33% of the respondents hired extra workforce for their agro-enterprises. Two (2%) of the agro-dealers employed part-timers, 29% had full-time employees, 1% employed both full-time and part-time workers, while the rest (67%) used family labour. A significant number (92%) of the agro-dealers reported having good access to market information. This includes information on prices, new products, competition, and market trends. Information access is often viewed to facilitate efficient decision-making in business and hence expected to positively influence the willingness to stock the new fungal-based biopesticide among the agro-dealers.

About 82% of the surveyed agro-dealers reported managerial support from their merchandise manufacturers and suppliers. Previous studies indicate that the availability of manpower in businesses triggers expansion of the ventures with a corresponding improved sale. This is because they are educated and trained on the new technologies to avoid obsolescence [30]. This is justified by Lin [31] on new product acceptance decisions. The availability of managerial support is therefore expected to positively influence the agro-dealers' willingness to stock the biopesticide. Social capital and networks are important for creating awareness and allowing the exchange of information about new products. About 44% of the agro-dealers participated in social networks (agro-dealer groups or cooperatives), which may positively influence their willingness to stock the new fungal biopesticide.

### 3.2. Agro-Dealers Knowledge of Biopesticides

Table 2 presents the agro-dealers' knowledge of biopesticides and other non-pesticide practices. Most (71%) of the agro-dealers were aware of non-chemical practices for the management of *T. absoluta*, while about 43% were aware of the biopesticide for suppression of the pest. Different cultural practices for management of the pest were reported by the respondents such as planting resistant tomato varieties, crop rotation with non-host crops, physical removal of the infested plant or plant parts among others (Table 2). Crop rotation was reported as the most popular cultural method for management of *T. absoluta* among the farmers that interact with the interviewed agro-dealers. The results corroborate with previous studies such as Retta and Berhe [32] who observed crop rotation with a host-free crop as an important cultural practice for eradication of tomato leaf miner.

**Table 2.** Agro-dealers' knowledge of biopesticides and other non-pesticide practices in Kirinyaga County.

| Management Method | Practice | Responses (% Yes) *n* = 141 |
|---|---|---|
| Non-synthetic chemical pesticides | non-chemical pesticides for control of Leaf miner | 70.92 |
| Non-pesticide practices | Knowledge on non-pesticide practices for control of Leaf miner (%yes) | 43.36 |
| Cultural practices | Planting resistant tomato varieties | 29.51 |
| | Selecting healthy seeds or sanitizing seed treatment | 19.67 |
| | Soil tillage | 13.11 |
| | Crop rotation with non-host crop | 78.69 |
| | Adjust planting/harvesting dates to reduce pest damage | 31.15 |
| | Adjust irrigation timing/amount to reduce pest damage | 37.70 |
| | Growing tomato under insect net or net house | 50.82 |
| | Picking and destroying infected plant or plant parts | 54.10 |
| | Using a barrier crop | 18.03 |
| Orchard sanitation | Collecting fallen infested tomatoes and burying underground | 54.10 |
| Trapping | Using Pheromones traps for monitoring, and mass trapping | 40.98 |
| | Hanging sticky traps | 16.39 |
| | Using water traps | 3.28 |
| Biological control | using parasitoids/natural enemies | 32.79 |

Source: Survey data.

Although orchard sanitation is also a cultural practice, we report it separately since it is one of the broader IPM practices that is recommended for suppression of *T. absoluta* in addition to the use of biopesticide among other supplementary strategies [33]. The importance of the practice is supported by the significant number of respondents (54%) who reported the management method in our study. Agro-dealers were also familiar with other tomato leaf miner IPM strategies such as the use of pheromone and water traps, as well as biological control using natural enemies.

### 3.3. Agro-Dealers' Perception towards Biopesticides in Kirinyaga County

The principal component analysis (PCA) method was used to extract perception indicators from agro-dealers. The robustness of the analysis was tested using Kaiser-Meyer-Olkin (KMO) test, Bartlett's test, and Cronbach's alpha tests. The KMO value of 0.8099 is higher than the accepted value of 0.5, which depicts that the sample was adequate and principal component analysis could be used for the given data set. The significance

of $p < 0.001$ acquired for Bartlett's test also indicates a sufficient measure of sampling adequacy. In addition, a Cronbach's alpha value of 0.7747 indicates a high level of internal consistency thus the indicators measured the same latent perception variables as shown in Table 3 below.

**Table 3.** Principal component analysis results with component loadings.

| | PC 1 Effectiveness in Crop Protection | PC 2 Effect on Health and Environment | PC 3 Commercialization Aspect | Unexplained |
|---|---|---|---|---|
| Use of Biopesticides in crop protection | 0.4676 | | | 0.2037 |
| Effectiveness of the biopesticides | 0.4916 | | | 0.1683 |
| Safe to humans and the environment | | 0.6635 | | 0.3463 |
| Willingness to trade | | 0.6139 | | 0.4176 |
| Contributes to high yield | 0.4807 | | | 0.1646 |
| Can replace synthetics | 0.4671 | | | 0.225 |
| Price when trading | | | 0.6046 | 0.4575 |
| Type of pesticides | | | 0.6426 | 0.4002 |
| Effect on health | | | | 0.6061 |
| Demand | | | 0.4319 | 0.6782 |
| Test for robustness | | | | |
| Kaiser-Meyer-Olkin | 0.8099 | | | |
| Bartlett's test | 0.0000 | | | |
| Cronbach's alpha | 0.7747 | | | |

We asked agro-dealers to rank their preference based on the use of biopesticides in crop protection, from 0 to 4 (0 for most preferred, 1 for preferred, 2 for less preferred, 3 for least preferred, and 4 for others). They were also asked if they consider pesticides characteristics such as price, the type of pesticide, the effect of pesticide on human health, and the pesticides demand when making stocking decisions. Further, they were also asked to rank their level of agreement with statements concerning biopesticides which included: 'biopesticides are effective in controlling *T. absoluta*', 'biopesticides are safe to both humans and the environment', willingness to trade biopesticides, 'biopesticides contribute to high yield as compared to synthetic pesticides', and if they believe that biopesticides can replace synthetic pesticides. After running the principal component analysis, three principal components were extracted from the agro-dealers' responses. The first one contributed the highest percentage (35%) to the variance, while the second and third components contributed 15% and 13%, respectively. The first principal component (PC 1) captured the following dominant factors; 'preference of biopesticide use in crop protection', 'biopesticides are effective in controlling *T. absoluta*', 'biopesticides contribute to high yield as compared to synthetic pesticides, and 'biopesticides can replace synthetic pesticides' (Table 3).

These results show that the PC 1's dominant factors were found to represent the effectiveness of biopesticides in crop protection over synthetic chemical pesticides. The dominant perception indicators captured in PC 2 were found to represent the general effect of biopesticides on human health and the environment, while PC 3 captured the aspect of commercialization and factors considered in decision making when trading in pesticides which included; 'price when trading', 'the type of pesticide' and 'the demand' as illustrated in Table 3.

### 3.4. Willingness to Stock Biopesticides among the Surveyed Agro-Dealers

Eighty-two (82%) percent of the interviewed agro-dealers were willing to stock the new ICIPE biopesticide at a price similar to the current buying price of the most used chemical pesticide for control of *T. absoluta* (Coragen®), estimated at Ksh. 622 (US$6.22)

per litre (Table 4). The rest rejected the offer. As noted earlier, the agro-dealers were asked the buying price for the most frequently sold pesticide for management of *T. absoluta*. This price was the first bid of the double-bounded contingent valuation, followed either by a higher or lower amount as explained earlier in the conceptual framework. They were then offered a discount ranging from 5% to 50% of their current cost of stocking the most used chemical pesticide for control of *T. absoluta*, randomly picked by the data collection program (CSPro) and presented to the agro-dealer. Once offered a discount, additional agro-dealers were willing to stock the fungal-based biopesticide. Those who accepted the first bid were then asked if they would be willing to stock the biopesticide at a higher price ranging from 5% to 50% of the initial bid. Additional agro-dealers were willing to stock at slightly above the initial bid. On average, the mean willingness to pay for the ICIPE 20 was Ksh. 1.018 per litre (US$10.18), about 64% over and above Coragen® market price. This suggests that the agro-dealers consider biopesticides more efficient than chemical pesticides such as Coragen®, and thus would be willing to stock and sell them to tomato growers (Table 4).

**Table 4.** The agro-dealers' mean willingness to pay and mean premium price for biopesticide and commonly used synthetic pesticide.

| Pesticide | Mean | Z Stat | % Change from Market Price |
|---|---|---|---|
| Fungal biopesticide (ICIPE 20) | 1018.19 | 17.21 *** | 64 |
| Coragen® | 621.96 | | |

Note: Statistical significance at 0.01 (***).

*3.5. Factors Affecting Willingness to Stock Fungal Based Biopesticide (ICIPE 20) in Kirinyaga County*

Table 5 presents the regression results of the factors influencing agro-dealers willingness to stock the fungal-based biopesticide (ICIPE 20). The age of the agro-dealers is negatively related to their willingness to stock the biopesticide. This suggests that the older the trader, the lower the willingness to pay for the biopesticide. The findings corroborate with Nyangau [11] and Bandara [34] who found that perception and willingness to pay for Bt 43 and ICIPE 78 in Uganda decreased with the age of the individuals. The education coefficient is positive and significantly related to the willingness to stock ICIPE 20. Education enhances the processing and interpretation of new information, thus better-educated agro-dealers may have a better understanding of the benefits of biopesticides. These results are consistent with the previous studies such as Constantine [28] who found that educated individuals are more likely to pay for biopesticides in Kenya.

**Table 5.** Factors influencing willingness to stock for a fungal-based Biopesticide (ICIPE 20) in Kirinyaga County.

| Variables | Coefficient | Standard Error | Z |
|---|---|---|---|
| Age of agro-dealers | −0.154 ** | 0.076 | −2.030 |
| Years of schooling of agro-dealers | 0.419 * | 0.246 | 1.700 |
| Agro-dealer access to social networks | 5.063 ** | 2.316 | 2.190 |
| Agro-dealer knowledge on None-pesticide practices | −1.985 | 1.296 | −1.530 |
| Years of business operation | −0.073 | 0.098 | −0.750 |
| Agro-dealer access to credit facilities | 2.408 * | 1.281 | 1.880 |
| Managerial support from manufacturers and suppliers | 1.477 | 0.996 | 1.480 |
| Availability of employees in the agro-vets | −1.352 | 1.124 | −1.200 |
| Agro-dealer access to information | 2.199 * | 1.233 | 1.780 |

Note: Dependent variable: (dummy variable: yes/no response to first bid); Statistical significance at 0.05 (**) and 0.1 (*).

Agro-dealers' access to social networks was similarly positive and significantly related to agro-dealers' willingness to stock the biopesticide. Access to social networks is expected to enhance agro-dealers' knowledgeable of the market trends including the availability of new products, prices, and consumer demand. The coefficient for access to credit facilities was also positive and significant at 10%, implying that agro-dealers with access to the credit facilities are more likely to accept new products such as the fungal-based biopesticide or increase their stocks in the stores due to the availability of capital. The findings are consistent with the previous studies such as Chia [29] who found that access to credit positively influenced willingness to pay for insects for feeds among agro-dealers in Kenya. Access to information was positive and significantly related to the willingness to stock the biopesticide, suggesting that agro-dealers with access to reliable information on both chemical and biopesticides such as their prices, quantity, and quality knew about them and their demand (Table 5). In the pesticides industry, information is thus very critical to both agro-dealers and the farmers.

## 4. Conclusions and Policy Implication

The main objective of this study was to determine the knowledge, perception, and willingness to stock a fungal-based biopesticide among agro-dealers in Kirinyaga County. Agro-dealers are indeed aware of the use of biopesticides in the management of *T. absoluta*. Similarly, they are aware of various cultural practices including orchard sanitation, as well as trapping innovations and use of natural enemies for suppression of the pest. Though the agro-dealers are well known for heavily stocking synthetic pesticides, they are aware of their adverse effects on human, animal, and environmental health, demonstrating the potential to influence the use of biopesticide as alternative crop protection to the use of hazardous chemical pesticides. The results further show that agro-dealers are willing to stock the biopesticide. Using the price of the most traded pesticide for management of *T. absoluta* (Coragen®), we find that 82% of the interviewed agro-dealers were willing to stock and retail at the same price the fungal-based biopesticide. On average the agro-dealers were willing to pay about 64% over and above Coragen®'s buying price, suggesting that the agro-dealers consider biopesticides effective in the management of *T. absoluta* and thus would be willing to stock and sell them to tomato farmers.

The empirical analysis of the study revealed that younger and educated agro-dealers are more likely to stock the new product, and thus potential influencers who can lead the awareness and thus the adoption of the biopesticide ICIPE 20 among tomato growers. Similarly, agro-dealers with access to credit, social networks, and information are more willing to stock the new fungal-based biopesticide.

Although we find useful insights into the feasibility of the commercialization of the ICIPE 20, we acknowledge limitations in our study. One was the use of the purposive sampling method. This implies that it was not possible to measure for variability. In addition, the results from the data cannot be generalized beyond the current sample and region of study. Second, our sample is relatively small, although it captured the whole population of agro-dealers in the study area. Future studies should consider other tomato-growing regions in Kenya and the larger SSA region. Besides, future research should incorporate the farmers are they are important actors in the biopesticide commercialization value chain.

Even with these limitations, the results suggest important policy implications. There is a need to increase the level of access to social networks among the agro-dealers for enhanced knowledge on pesticide types, quality, and quantity. Credit facilities and information access among agro-dealers are also vital since they are critical factors that easily influence the knowledge, perception, and willingness to stock new products. There is merit to commercialize ICIPE 20 for use as an integral component of IPM in the management of the devastating invasive South American tomato leaf miner *T. absoluta*.

**Author Contributions:** B.W.M., S.A.M., S.N. and F.M.K. conceptualized and drafted the proposal for the study. F.O., P.M.M. and B.W.M. were involved in developing the methodology and formal

data analysis. The text was written collaboratively by all co-authors. The published version of the manuscript has been read and approved by all authors. All authors have read and agreed to the published version of the manuscript.

**Funding:** This research was funded by Biovision Foundation Project (grant No.: BV DPP-012/2019-2022), and the African Union (AU) (grant no.: AURG II-2-123-2018). The first author is grateful to the Centre of Excellence in Sustainable Agriculture and Agribusiness Management (CESAAM), for the partial support in pursuing his Master's program at Egerton University. We also acknowledge the International Centre of Insect Physiology and Ecology's core funding from the Swedish International Development Cooperation Agency (Sida); the Swiss Agency for Development and Cooperation (SDC); the Federal Democratic Republic of Ethiopia; and the Government of the Republic of Kenya. The views expressed herein do not necessarily reflect the official opinion of the donors.

**Institutional Review Board Statement:** Ethical assessment and authorization for this project were granted as a result of the International Centre for Insect Physiology and Ecology's substantial work on a range of insect pests at the study site. However, survey respondents' oral agreement was obtained after they had been made aware of the study to make informed and free decisions about their participation.

**Informed Consent Statement:** Informed consent was obtained from all subjects involved in the study.

**Data Availability Statement:** The data presented in this study are available on request from the corresponding author.

**Acknowledgments:** The authors would like to express their gratitude to Kirinyaga county agro-dealers who volunteered their time to participate in the survey and the enumerators for their effort in data collection. An earlier version of this paper was presented at the 31st International Conference of Agricultural Economists, 17–31 August 2021, Online. We acknowledge the insightful comments from participants of this conference.

**Conflicts of Interest:** The authors declare no conflict of interest.

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
