# Peer review of "Agro-Dealers’ Knowledge, Perception, and Willingness to Stock a Fungal-Based Biopesticide (ICIPE 20) for Management of Tuta absoluta in Kenya"

_agriculture, doi:10.3390/agriculture12020180_

Round 1
Reviewer 1 Report
My overall impression of the paper is positive. It is an informative and well-performed study. Some improvements, however, could potentially increase the readability and contribution of the study to the literature:
- The Introduction says that the "study seeks to contribute to the existing literature on the commercialization and adoption of new agricultural technology" (lines 86-87). However, the discussion of these gaps in previous literature on commercialization and technologies is rather weak. In the Introduction, the author concentrates on the situation in Kenya which is good for a practice-oriented part of the Introduction. However, the theoretical relevance of the topic should also be addressed. My recommendation is to review the literature, identify major gaps, and demonstrate how the paper attempts to address them. Theoretical relevance of the topic should be emphasized and generalized for other countries, not only Kenya. This would increase the value of the paper.
- In lines 93-100, the author summarizes the results. I think the Introduction section is not a proper place for such a summary. Consider moving it to the Conclusion.
- In the abstract, try to avoid presenting too detailed results, such as percentages and prices. Instead, summarize tendencies and focus on explaining the relevance of the study (overall relevance and specific relevance for Kenya), methods, and potential theoretical (now missing) and practical implications of the findings
Author Response
Responses to reviewer’s comments for paper
“Agro-dealers’ knowledge, perception, and willingness to stock a fungal based biopesticide (ICIPE 20) for management of Tuta absoluta in Kenya.”
We appreciate you taking the time to read our work and providing us with constructive criticism so that we could improve it. Below are our responses to your comments and suggestions. We sincerely hope you are pleased with our responses.
Response to Reviewer 1 Comments
- Point 1: My overall impression of the paper is positive. It is an informative and well-performed study. Some improvements, however, could potentially increase the readability and contribution of the study to the literature
Response 1: Thank you for the positive feedback on our manuscript. We have below addressed your concerns and suggestions.
- The Introduction says that the "study seeks to contribute to the existing literature on the commercialization and adoption of new agricultural technology" (lines 86-87). However, the discussion of these gaps in previous literature on commercialization and technologies is rather weak. In the Introduction, the author concentrates on the situation in Kenya which is good for a practice-oriented part of the Introduction. However, the theoretical relevance of the topic should also be addressed. My recommendation is to review the literature, identify major gaps, and demonstrate how the paper attempts to address them. Theoretical relevance of the topic should be emphasized and generalized for other countries, not only Kenya. This would increase the value of the paper.
Response 3: Thank you for this feedback. We have edited this section. Unfortunately, due to space limitations, we are not able to include a comprehensive literature review on the topic; however, we have cited a few studies that relate to this work (see the fourth paragraph in the introduction).
- Point 3: In lines 93-100, the author summarizes the results. I think the Introduction section is not a proper place for such a summary. Consider moving it to the Conclusion
Response 3: We appreciate you bringing this to our attention. We looked at a few articles that have been published in this journal that have a structure that is similar to the introduction. Due to the nature of our research objectives, we would prefer to keep the results in the introduction so that our readers are aware of what to expect later in the main result part.
- Point 4: In the abstract, try to avoid presenting too detailed results, such as percentages and prices. Instead, summarize tendencies and focus on explaining the relevance of the study (overall relevance and specific relevance for Kenya), methods, and potential theoretical (now missing) and practical implications of the findings
Response 4: Thank you for your comment and informative suggestion. We have edited the abstract and hope we have adequately captured the missing bits.

Reviewer 2 Report
Summary:
The study is dedicated to pest management and agrodealers willingness to stock up on a fungal-based biopesticide to manage a leaf miner causing damage in Kenyan tomato production. The study builds on a bidding game in the contingent valuation method, more precisely a double-bounded model and a binary regression model. The results show that a higher proportion of agro-dealers (82%) were willing to pay for the bio-insectice at the same price as Coragen®, a widely used insecticide at a price of approximately Ksh. 750 (US$ 7.5) per liter. The regression analysis revealed that individual characteristics such as age, education, access to social
networks and to credit facilities, and information are associated with the willingness to stock the biopesticide.
I enjoyed reading the paper however I have a few concerns
The abstract indicates that this paper serves as a case study. However any information related to the authors approach to case study is completely missing. On the background of the chosen research aim and method, I would shy away from case study claims. If you would like to keep it you need to considerably elaborate to section 2.2 and add information and justify your approach to case study, the school you follow and how your analysis fits in with these aspects
The study use purposive sampling. Could the authors acknowledge limitation of this sampling approach in the conclusion of the study. Moreover I was wondering if the chosen agrodealers received a small token of appreciation ( time compensation) or which incentives where involved to participate in the study.
I have further a question about the questionaire and the work with the agrodealers and enumorators. Was the questionaire written in English language or in a local language. If translation was involved how did you assure translation accuracy.
Who were the enumerators and who trained them? Was it students trained by the researchers, or individuals specifically employed for cencus enumeration? Please elaborate on these information
A double-bounded model has been used and the authors argue for the appropiateness of this model, by explaining the disadvantages of single bounded model and the multi-bounded model. While the explanation of the single bounded model makes sense to me. I would like to see a better justification why the multiple bounded model was not used, given the possibility to involve multple bids, and incoperate uncertainty. That sounds to me allowing for a more realistic scenario. I understand the argument of design bias, but surely there must be ways to overcome this. So my questions: Have authors tried to use a multiple bound model and safeguard against design bias? Or why is the double bound model still most appropiate with respect to the kenyan agrodealer reality. Otherwise please acknowledge limitation in the conclusion section.
Please add to the conclusion section suggestions for future research, this seems to be missing.
One major concern: Please make sure that you do not self-plagiarize and rewrite section that are exactly the same as in your conference paper. See Ogutu, F. (2021). Agro-dealer’s knowledge, perception, and willingness to stock a fungal based biopesticide (ICIPE 20) for management of Tuta absoluta in Kenya presented at the ICAE 2021,
Author Response
Response to Reviewer 2 Comments
Point 1: I enjoyed reading the paper however I have a few concerns
The abstract indicates that this paper serves as a case study. However, any information related to the authors approach to case study is completely missing. On the background of the chosen research aim and method, I would shy away from case study claims. If you would like to keep it you need to considerably elaborate to section 2.2 and add information and justify your approach to case study, the school you follow and how your analysis fits in with these aspects
Response 1: Thank you for the positive response and the suggestions. This was indeed an oversight, with the “case” here referring to the study area. We have edited this to read “Using Kirinyaga county where tomato production is predominant”.
Point 2: The study use purposive sampling. Could the authors acknowledge limitation of this sampling approach in the conclusion of the study. Moreover, I was wondering if the chosen agrodealers received a small token of appreciation (time compensation) or which incentives were involved to participate in the study.
Response 2: Thank you for bringing this to our attention. We have now acknowledged the limitations of the purposive sampling technique in the conclusion section (see the second last paragraph of the conclusion).
Regarding the incentives, although we asked for voluntary participation, for this study we offered a small non-monetary incentive which was only revealed at the end of the interview to avoid any bias of the responses provided by the trader.
Point 3: I have further a question about the questionnaire and the work with the agrodealers and enumerators. Was the questionnaire written in English language or in a local language. If translation was involved, how did you assure translation accuracy.
Response 3: Thank you for this question. The questionnaire was written in English and comprehensively explained in the local language (the national language Swahili and the local language of the study area) during the training. This was done to ensure the enumerators had the same understanding of the questions, and thereafter would administer them in the same way to the respondents. We have added a sentence in 2.2 to reflect this.
Point 4: Who were the enumerators and who trained them? Was it students trained by the researchers, or individuals specifically employed for census enumeration? Please elaborate on these information
Response 4: This is a valid concern. The enumerators were trained for four days before pretesting the questionnaire, then proceeded to collect the data. The training was conducted by the lead author of this manuscript, who also supervised the data collection exercise. The enumerators were post-graduate students with vast experience in data collection and most of them spoke the local language of the study area. We have added a sentence to illustrate this in section 2.2.
Point 5: A double-bounded model has been used and the authors argue for the appropriateness of this model, by explaining the disadvantages of single bounded model and the multi-bounded model. While the explanation of the single bounded model makes sense to me. I would like to see a better justification why the multiple bounded models was not used, given the possibility to involve multiple bids, and incorporate uncertainty. That sounds to me allowing for a more realistic scenario. I understand the argument of design bias, but surely there must be ways to overcome this. So my questions: Have authors tried to use a multiple bound model and safeguard against design bias? Or why is the double bound model still most appropriate with respect to the Kenyan agrodealer reality. Otherwise please acknowledge limitation in the conclusion section.
Response 5: Thank you for the comment and suggestion. We have added additional drawbacks to the use of multiple-bounded methods in section 2.1 (see last few sentences of the second paragraph)
Point 6: Please add to the conclusion section suggestions for future research, this seems to be missing.
Response 6: Thank you. We have added this in the conclusion section as suggested.
Point 7: One major concern: Please make sure that you do not self-plagiarize and rewrite section that are exactly the same as in your conference paper. See Ogutu, F. (2021). Agro-dealer’s knowledge, perception, and willingness to stock a fungal based biopesticide (ICIPE 20) for management of Tuta absoluta in Kenya presented at the ICAE 2021,
Response 7: Thank you for raising this concern. We are sorry for the oversight; we forgot to mention in the acknowledgment, which we have now done. A previous version of this paper (as you correctly indicated) was presented at the ICAE, and it is contained in the conference proceedings. Accordingly, the paper appears in the conference proceedings under AgEcon Search. We hope that the journal doesn’t consider this as a conflict, as also indicated by the AgEco search managers and International Association of Agricultural Economists (IAAE) president “Inclusion of a conference paper in AgEcon Search is not considered publication. It honors the long-time tradition of economists producing working papers and conference papers on a project before any journal article submissions”. They further refer to the IAAE’s journal Agricultural Economics https://onlinelibrary.wiley.com/journal/15740862 as a good example of how economics journals view working papers and conference papers. We hope our explanation ameliorates the above concern. Thank you.

Round 2
Reviewer 1 Report
My recommendations have been addressed
Reviewer 2 Report
The review significantly improved the paper. All my concerns have been adressed. Looking forward to see the paper.